# GLI3 resides at the intersection of hedgehog and androgen action to promote male sex differentiation

Anbarasi Kothandapani[1], Samantha R. Lewis[1], Jessica L. Noel[1], Abbey Zacharski[1☯‡], Kyle Krellwitz[1☯‡], Anna Baines[1☯‡], Stephanie Winske[1‡], Chad M. Vezina[1], Elena M. Kaftanovskaya[2], Alexander I. Agoulnik[2], Emily M. Merton[3], Martin J. Cohn[3], Joan S. Jorgensen[1]*

**1** Department of Comparative Biosciences, University of Wisconsin-Madison, Madison, Wisconsin, United States of America, **2** Department of Human and Molecular Genetics, Florida International University, Miami, Florida, United States of America, **3** Molecular Genetics and Microbiology, University of Florida, Gainesville, Florida, United States of America

☯ These authors contributed equally to this work.
‡ Undergraduate Researchers
* joan.jorgensen@wisc.edu

**Data Availability Statement:** All relevant data are within the manuscript and its Supporting Information files.

## Abstract

Urogenital tract abnormalities are among the most common congenital defects in humans. Male urogenital development requires Hedgehog-GLI signaling and testicular hormones, but how these pathways interact is unclear. We found that $Gli3^{XtJ}$ mutant mice exhibit cryptorchidism and hypospadias due to local effects of GLI3 loss and systemic effects of testicular hormone deficiency. Fetal Leydig cells, the sole source of these hormones in developing testis, were reduced in numbers in $Gli3^{XtJ}$ testes, and their functional identity diminished over time. Androgen supplementation partially rescued testicular descent but not hypospadias in $Gli3^{XtJ}$ mutants, decoupling local effects of GLI3 loss from systemic effects of androgen insufficiency. Reintroduction of GLI3 activator (GLI3A) into $Gli3^{XtJ}$ testes restored expression of Hedgehog pathway and steroidogenic genes. Together, our results show a novel function for the activated form of GLI3 that translates Hedgehog signals to reinforce fetal Leydig cell identity and stimulate timely INSL3 and testosterone synthesis in the developing testis. In turn, exquisite timing and concentrations of testosterone are required to work alongside local GLI3 activity to control development of a functionally integrated male urogenital tract.

## Author summary

Disorders in male sex differentiation (DSD) are among the most common defects in all live births, yet in many cases, pediatric patient families are reluctant to address the issue and endure lifelong consequences. Urogenital tract development, as in many organ systems, depends on exquisite timing among layers of a number of signaling pathways. Here, we show that interactions between the hedgehog and androgen signaling pathways are required for the development of internal and external male sex characteristics, but results

**Funding:** "Funding sources include the National Institute of Health, The Eunice Kennedy Shriver National Institute of Child Health and Human Development R01HD075079 (JSJ), R01-HD090660 (JSJ) and R01-HD090660-S (AK, JSJ); NIH, National Institute of Arthritis and Musculoskeletal and Skin Diseases R01-AR070093 (AIA); NIH, National Institute of Diabetes and Digestive and Kidney Diseases R01-DK110408 (MJC) and U01-DK110807 (CMV) and the University of Wisconsin-Madison (JSJ). The funders had no role in study design, data collection and analysis, decision to publish, or preparation of the manuscript."

**Competing interests:** The authors have declared that no competing interests exist.

for each tissue is distinct. This new knowledge will aid in discovering the means by which congenital malformations might occur, identify potential developmental targets that might be vulnerable to environmental exposures, and promote new ideas for how they might be prevented.

## Introduction

Congenital anomalies within the male urogenital tract, including cryptorchidism (failure of the testes to descend) and hypospadias (mislocalization of the urethral meatus), are among the most frequently documented abnormalities in all live births and their incidence is increasing [1]. These defects in male sex differentiation can occur separately or together with additional male-specific disorders suggesting a common or overlapping etiology. It has long been established that male sex differentiation requires androgen activity that must occur within a critical timeframe of fetal development [2]. Indeed, there is concern that the overall increase in the incidence of disorders of male sex differentiation that has been observed in developed industrial countries is related to endocrine disruptors that impact androgen activity [3]. Notably, a number of other molecular pathways, including hedgehog morphogens, are also known to control aspects of development that define male primary and secondary sex organ phenotypes [4–6].

Sonic Hedgehog (SHH) directs patterning and development of external genitalia and secondary sex organs while Desert Hedgehog (DHH) stimulates differentiation of a critical population of cells within the developing testis, the fetal Leydig cells, that are the sole source for fetal androgen production [5,7,8]. All hedgehog ligands signal via Smoothened receptor to activate GLI1, GLI2 and GLI3 transcription factors. Each factor has been shown to contribute to male sex differentiation in distinct ways. For example, while mutations in *Gli2* alone caused external genitalia defects, compound mutation with *Gli1* showed no additional defects, but its combination with *Gli3* exhibited more severe defects [5,9–11]. Further, some studies showed that *Gli3* is not essential for the development of external genitalia but others found that *Gli3* disruption causes significant genital malformations [5,10,11]. Meanwhile, within the fetal testis, elimination of *Gli1* or *Gli2* individually is not sufficient to reproduce the failure in fetal Leydig cell differentiation seen in *Dhh* knockout mice; *Gli3* has not been tested [12]. Together, these studies support the concept that hedgehog and androgen pathways collaborate to orchestrate male sex differentiation and highlight *Gli3* in particular as a potentially important intermediary between the two. In humans, there is a subset of male patients with mutations in *GLI3*, including autosomal dominant *GLI3* anomalies categorized as Greig cephalopolysyndactyly syndrome (GCPS) and Pallister-Hall syndrome (PHS), that exhibit differences in male sex differentiation [13,14]. Despite these reports, the role for GLI3 in urogenital development remains unknown.

In this report, we used the *Gli3* extra toes Jackson (*XtJ*) mouse model that harbors a loss-of-function mutation and is an established model for human GCPS [15]. Our results show that, similar to GCPS patients, the *Gli3$^{XtJ}$* mutation causes differences in male sex differentiation including cryptorchidism and hypospadias. Notably, testicular hormones, INSL3 and testosterone, were significantly lower in *Gli3$^{XtJ}$* mutant male embryos, which contributed to a range of mutant phenotypes in primary and secondary organs of the male reproductive tract. While testis development, including testis cord formation, progressed normally, expression of genes associated with androgen and INSL3 production in addition to members of the DHH pathway were disrupted within *Gli3$^{XtJ}$* mutant testes, indicating a failure in fetal Leydig cell differentiation or maintenance. Reintroduction of a constitutively active *Gli3*-full length expression plasmid (GLI3A) into *Gli3$^{XtJ}$* mutant testes significantly increased fetal Leydig cell-specific gene

expression providing evidence that GLI3A maintains fetal Leydig cell identity. Together, our results suggest that fetal Leydig cell function depends on activated GLI3 for optimal expression of genes critical to INSL3 and androgen synthesis, the timing and concentration of which is critical to define optimal male sex differentiation.

## Results

### The *Gli3^{XtJ}* mutation disrupts male sex differentiation

*GLI3* mutations have been associated with the urogenital phenotypes exhibited by GCPS and PHS patients [13,14]. Thus, we used *Gli3^{XtJ}* mutant mice, which serve as a robust model of GCPS [15], to investigate the interplay between *Gli3* and androgen production and action. While mice heterozygous for the *Gli3^{XtJ}* mutation were viable and fertile, the homozygous mutation resulted in embryonic lethality, thus confining our analysis to fetal development up to embryonic day (E) E17.5 [15]. Similar to some GCPS patients, *Gli3^{XtJ}* mutant male embryos exhibited cryptorchidism and external genitalia defects, including hypospadias. While wild type testes were present near the bladder by E16.5, all *Gli3^{XtJ}* mutant males examined exhibited cryptorchidism with placement of the testes near the kidney (Fig 1A and 1B). This difference was quantified by measuring the distance from the kidney hilum to the center of the testis on the ipsilateral side. Mutant testes were significantly closer to the kidneys than wild type testes, indicating a failure to descend. As a control for generalized developmental delays, caudal kidney to cranial bladder distances were measured and revealed no differences between wild type and mutant animals (Fig 1C). By E17.5 however, testes were located near the bladder in most *Gli3^{XtJ}* mutant male embryos examined, indicating that the transabdominal phase of testis descent may be variable and delayed by ~24 hours.

Next, examination of external genitalia using light microscopy (Fig 1D and 1E) and nanoscale computed tomography (Fig 1F and 1G) revealed abnormal penile morphology with varying degrees of hypospadias in *Gli3^{XtJ}* mutant male embryos (Fig 1E and 1G). External genitalia were examined in 6 litters of mice collected at E16.5–17.5 from which 6 male homozygous *Gli3^{XtJ}* mutants were recovered. Hypospadias was detected in half of the male mutants (n = 3/6), all of which showed ectopic opening of the urethral duct at the base of the penis (Fig 1E and 1G). In one case, the hypospadiac urethral opening extended from the base of the penis into the scrotum, similar to human penoscrotal hypospadias. Development of the penile glans and prepuce was generally normal except at the site of the ectopic opening.

Additional male sex phenotype characteristics arise starting around E16 and are related to hedgehog and androgen activity, including prostate bud formation and expansion of the anogenital distance (AGD) [16,17]. To investigate prostate bud initiation, we performed whole mount *in situ* hybridization on urogenital sinuses at E16.5 with a RNA probe for *Nkx3-1*, a prostate epithelium selective marker. Results showed no difference in bud patterns between wild type and *Gli3^{XtJ}* mutant urogenital tracts (Fig 1H and 1I). At the same time, however, *Gli3^{XtJ}* mutant male embryos showed a significantly reduced AGD compared to wild type littermates (Fig 1J). Altogether, the spectrum of phenotypes in male reproductive organs in *Gli3^{XtJ}* mutant embryos ranged from unaffected prostate bud formation to varying degrees of severity of AGD expansion, hypospadias, and cryptorchidism.

### Insufficient hormone production from fetal testes in *Gli3^{XtJ}* mutant embryos causes a subset of differences in male sex differentiation

The fetal testis is known to be the sole source of the two hormones that are critical for male sex differentiation: insulin-like 3 peptide (INSL3) and testosterone. To test whether the *Gli3^{XtJ}*

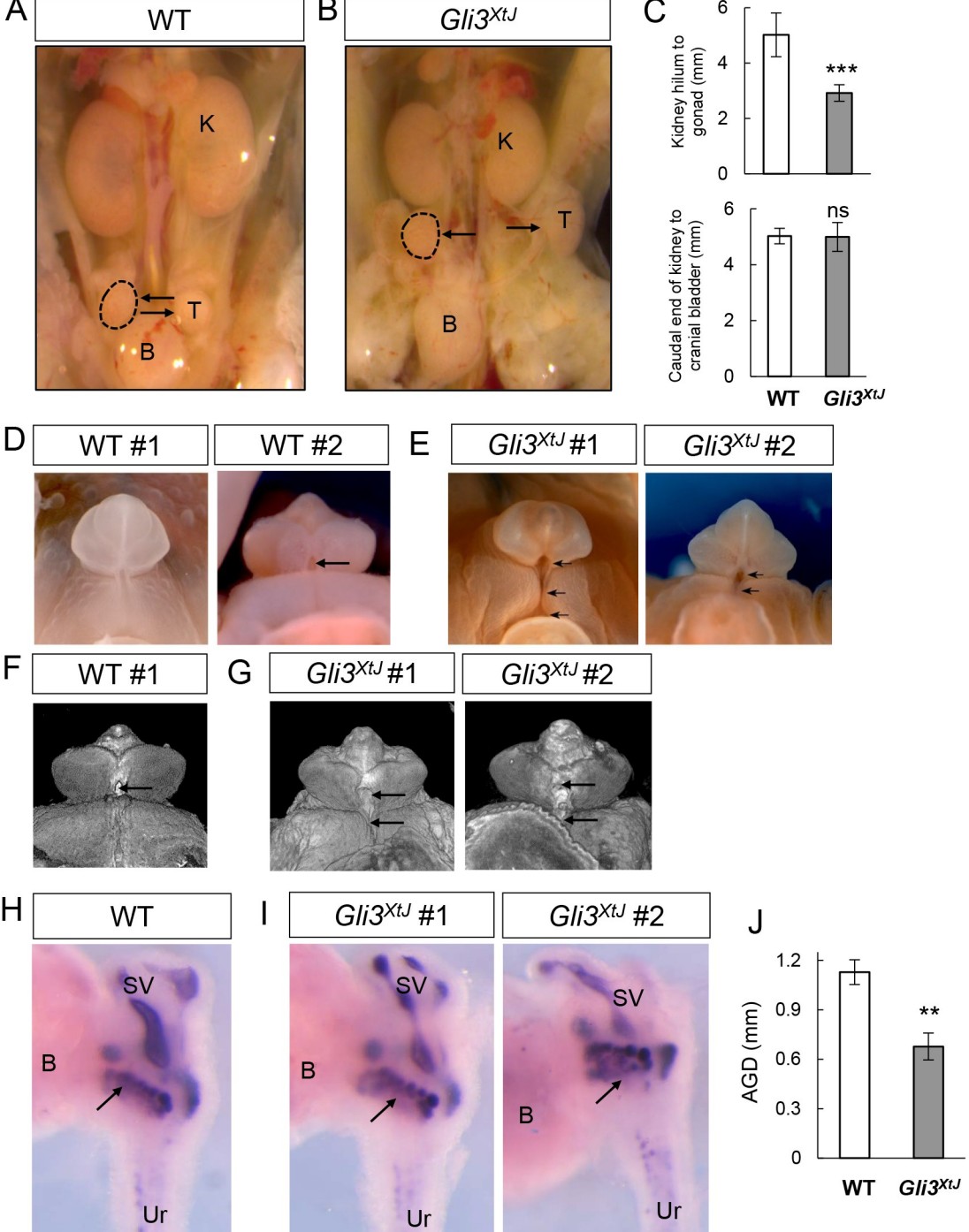

**Fig 1. *Gli3^{XtJ}* mutation disrupts male sex differentiation.** Wild type and *Gli3^{XtJ}* urogenital tracts were evaluated at E16.5 or E17.5 for testis placement (A,B), external genitalia (D-G), prostate bud formation (H, I) and anogenital distance (J). **A,B)** Wild type testes (n = 7) appear near the bladder (A) and *Gli3^{XtJ}* testes (n = 5) are located near the kidney showing bilateral cryptorchidism (B). Arrows denote testes (outlined with dashed line or T), kidney (K) and bladder (B). **C)** Distances from kidney hilum to gonad on ipsilateral side, and from the caudal kidney to the cranial bladder were measured. **D-G)** External genitalia in wild type (D,F) (n = 6) and *Gli3^{XtJ}* mutant male embryos (E,G) (n = 6) imaged in whole-mount using light microscopy (D,E) or nanoscale computed tomography (F,G). Single arrows mark the position of the urethral duct; multiple arrows indicate ectopic opening of hypospadiac urethra. **H, I)** Whole-mount male urogenital sinus was stained by ISH to detect *Nkx3-1* mRNA to visualize prostate buds in wild type (H) (n = 3) and *Gli3^{XtJ}* embryos (I) (n = 3). Seminal vesicles (SV), urethra (Ur), bladder (B), arrow denotes prostate buds. **J)** *Gli3^{XtJ}* embryos (n = 7) showed reduced

anogenital distance (AGD) compared to wild type embryos (n = 7). For C and J, results are represented as mean ± SEM. Student's t-test, **: p<0.01, ***: p<0.001, ns: non-significant.

mutation altered fetal testis hormone production, we measured *Insl3* transcript levels, which function as a readout for the INSL3 peptide [18], and testicular testosterone concentrations (Fig 2). *Insl3* transcripts in *Gli3$^{XtJ}$* mutant testes were significantly decreased by 64% at E13.5 and by 60% at E16.5 compared to wild type values; however, expression of *Insl3* increased by ~3-fold between E13.5–16.5 in both wild type and *Gli3$^{XtJ}$* mutant testes (Fig 2A). In addition, testicular testosterone levels were ~60% and over 70% lower in *Gli3$^{XtJ}$* testes compared to that of wild type testes at E13.5 and E16.5, respectively. Notably, in contrast to *Insl3*, *Gli3$^{XtJ}$* mutant testes failed to produce any additional testosterone while there was a ~2-fold increase in wild type testes between E13.5–16.5 (Fig 2B).

Early development of mouse external genitalia is controlled by locally acting signals, including SHH, in the absence of hormonal signaling [6,19,20], followed by androgen mediated virilization of the penis [21]. To test whether defects in *Gli3$^{XtJ}$* mutant male external genitalia were caused by the deficiency in androgens, we administered 1mg/kg DHT to pregnant dams daily starting at E12.5 and harvested embryos at E16.5 and E17.5. Proximal hypospadias was observed in 8 of 10 *Gli3$^{XtJ}$* mutant males treated with DHT, and the range of severity was similar to that observed in untreated and vehicle (sesame oil) treated *Gli3$^{XtJ}$* mutants (S1 Fig; compare S1B Fig with Fig 1E and 1G). Thus, androgen supplementation is not sufficient to restore normal urethral tube development in the absence of *Gli3*.

Testis descent is completed in two phases: transabdominal and inguinoscrotal, which occur pre- and postnatally, respectively, in the mouse [22]. During fetal stages, the urogenital ridges are held in their abdominal position by two ligamentous structures, the cranial suspensory ligament (CSL) and caudal gubernaculum. Because *Gli3$^{XtJ}$* mutant embryos are embryonic lethal, we were restricted to analysis of the transabdominal phase of testis descent that requires both INSL3 and testosterone, which promote outgrowth of the gubernaculum and regression of the CSL, respectively [23,24]. Both ligaments were evaluated by serial section histology at E16.5. Results show that there is no difference in morphology of the gubernaculum between wild type and mutant animals (Fig 3A). To test whether the hedgehog pathway has a direct impact on gubernaculum outgrowth, we stained *Gli1-LacZ* embryos that were collected at birth as a readout of active hedgehog signaling (S1 Methods). Results show positive β-galactosidase activity within the testes, bladder, and vas deferens as expected [25]. In contrast, staining in the gubernaculum was absent, suggesting that its development is independent of hedgehog activity (S2A Fig). INSL3 signaling is mediated through its cognate Relaxin Family Peptide Receptor type 2 (RXFP2) expressed in the gubernaculum [26–28]. qPCR results indicated that *Rxfp2* transcripts were not different between wild type and *Gli3$^{XtJ}$* mutant gubernacula (S2B Fig). Together, these data suggest that although the gubernaculum structure is normal, insufficient INSL3 exposure may contribute to delayed testicular descent in *Gli3$^{XtJ}$* mutants.

Additional evaluation of histological serial sections in wild type animals show typical regression of the CSL; however, the ligament remained intact and appeared shorter and thicker in *Gli3$^{XtJ}$* mutant animals (Fig 3A). Regression of the CSL depends on androgen/androgen receptor (AR) interactions [24,26,27,29]; therefore, we tested whether abnormal AR expression contributed to impaired CSL degradation (S1 Methods). Immunohistochemistry results showed no difference in AR expression in the CSL, epididymis, or testis between wild type and *Gli3$^{XtJ}$* mutant embryos (S2C Fig). Next, we evaluated testis descent following gestational exposure to DHT or sesame oil control. Although prenatal DHT treatment did not hasten descent compared to vehicle treated mutant embryos, fluorescent imaging of picrosirius red

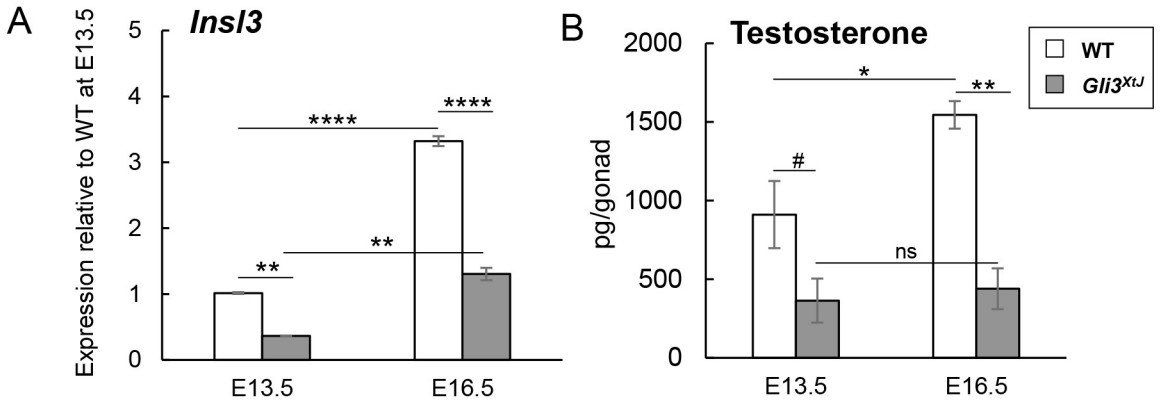

**Fig 2. *Gli3^XtJ* mutant testes produce insufficient hormones. A**) RT-qPCR comparing expression of *Insl3* transcripts in wild type (n = 3) and *Gli3^XtJ* (n = 3) testes at E13.5 and E16.5. Results are reported relative to E13.5 wild type testis. **B**) Testicular testosterone levels as measured by ELISA at E13.5 and E16.5 (WT n = 6; *Gli3^XtJ* n = 4). Results are represented as mean ± SEM. (A, B) One-way ANOVA followed by post-hoc Tukey's test, #: p<0.1, *: p<0.05, **: p<0.01, ****: p<0.0001, ns: non-significant.

staining did reveal that the collagen content of the CSL was affected (Fig 3B). CT-FIRE software was used to quantify individual collagen fiber metrics and total collagen density [30,31]. Collagen density per area was significantly decreased within the CSL of DHT treated compared to oil treated *Gli3^XtJ* mutant embryos (Fig 3C). In summary, these data suggest that insufficient production of testicular INSL3 and testosterone caused the delayed testis descent in *Gli3^XtJ* mutants.

### Inadequate testis hormone production is caused by differentiation of fewer fetal Leydig cells within an otherwise normal testis in *Gli3^XtJ* mutant embryos

Insufficient production of testicular hormones by *Gli3^XtJ* mutant testes imply that GLI3 functions as a mediator for DHH-induced fetal Leydig cell differentiation. To localize GLI3 presence in fetal testis, we performed double label immunohistochemistry for β-galactosidase and 3-βhydroxysteroid dehydrogenase (3βHSD), a fetal Leydig cell specific marker, on E16.5 testis sections from *Gli3*-LacZ reporter mice. As expected, *Gli3* expression was confined to the testis interstitium, specifically within peritubular myoid and fetal Leydig cells (Fig 4A).

We next considered the impact of the *Gli3^XtJ* mutation on testis development. Histological analysis of E13.5 and E16.5 testes showed intact testicular morphogenesis in both wild type and *Gli3^XtJ* mutant animals (Fig 4B–4E). Wild type and *Gli3^XtJ* mutant testes exhibited similar anti-Müllerian hormone (AMH) and Tra98 staining patterns suggesting no difference in number of Sertoli or germ cells, respectively (Fig 4F–4I). Laminin staining of the basement membrane, a product of Sertoli and peritubular myoid cells, was intact in *Gli3^XtJ* mutants indicating no impairment in function of either cell type (Fig 4J–4M). Together, these results suggest that the *Gli3^XtJ* mutation does not impact Sertoli, germ cell, or peritubular myoid cells, which together promote normal testis cord development. Given the decreased levels of fetal Leydig cell derived hormones, we next performed immunohistochemistry for 3βHSD, a fetal Leydig cell marker, to evaluate and quantify differentiated cells. Results show that fetal Leydig cells were present but fewer in number at both E13.5 and E16.5 in *Gli3^XtJ* mutant testes; however, fetal Leydig cell number/area increased at the same rate for both groups between the time points (Fig 5A–5F).

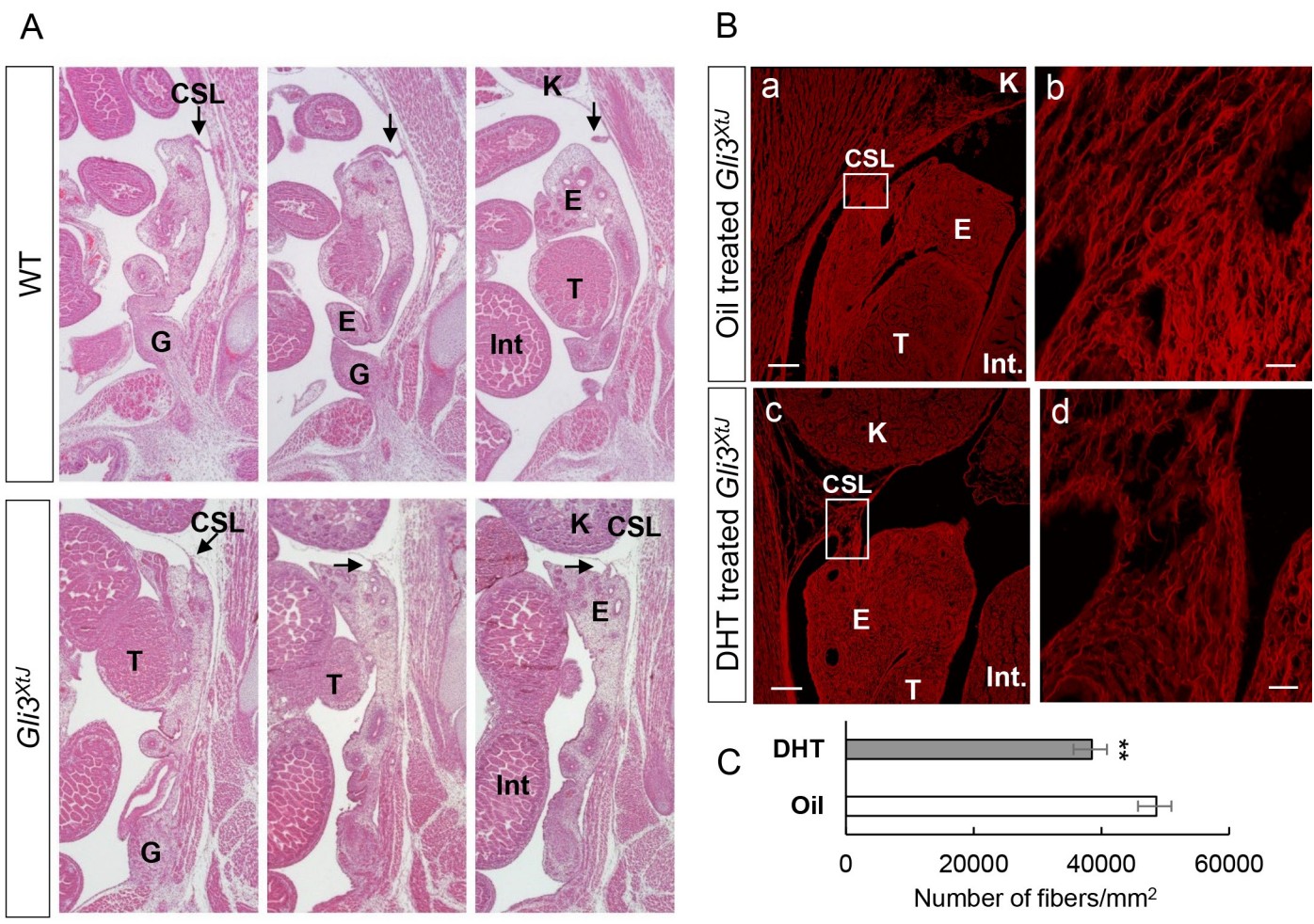

**Fig 3. Cranial suspensory ligaments (CSL) persist in *Gli3^XtJ* embryos and degrade in response to exogenous DHT. A)** H&E stain of serial sagittal sections from wild type (n = 5) and *Gli3^XtJ* mutant (n = 5) embryos at E16.5. Arrows denote cranial suspensory ligaments (CSL), epididymis (E), intestine (Int), kidney (K), testis (T) and gubernaculum (G). **B)** Pregnant dams were treated daily with sesame oil or DHT (1 mg/kg body weight), starting on E12.5 until the embryos were harvested at E16.5. Oil treated (a,b) (n = 4) and DHT treated (c,d) (n = 4) *Gli3^XtJ* urogenital sections were stained with picrosirius red to detect mature collagen fibers. The enlarged images (b,d) are taken from boxed regions in low magnification image (a,c). Scale bar: (a,c) = 100 μm and (b,d) = 25 μm. **C)** Total collagen density was determined by measuring the number of collagen fibers within a region of interest representing the total cross-sectional area of the CSL. Results are represented as mean ± SEM. Student's t-test, **: p<0.01.

Fetal Leydig cells are mitotically inactive and rarely proliferate [32]; therefore, the decrease in fetal Leydig cell numbers could be due to an increase in Leydig cell death or increased accumulation of progenitor cells that fail to differentiate [33]. Expression of cleaved caspase-3 was absent in both wild type and *Gli3^XtJ* mutant testes ruling out caspase 3-mediated fetal Leydig cell apoptosis (S3A Fig). To test for impaired fetal Leydig cell differentiation, we investigated expression of progenitor population markers and evaluated the integrity of signaling pathways that collaborate with DHH. Immunofluorescence and qPCR results show that the expression patterns for fetal somatic cell progenitor, WT1 [34,35] and adult Leydig cell progenitor, NR2F2 (COUPTF2) [36] are not different between wild type and *Gli3^XtJ* mutant testes (S3B and S3C Fig). Finally, it has been reported that future fetal Leydig cells differentiate in response to coordinated events between active DHH and PDGFA signals and a loss of the NOTCH pathway [37,38]. Transcript levels of PDGFA pathway genes and NOTCH family members

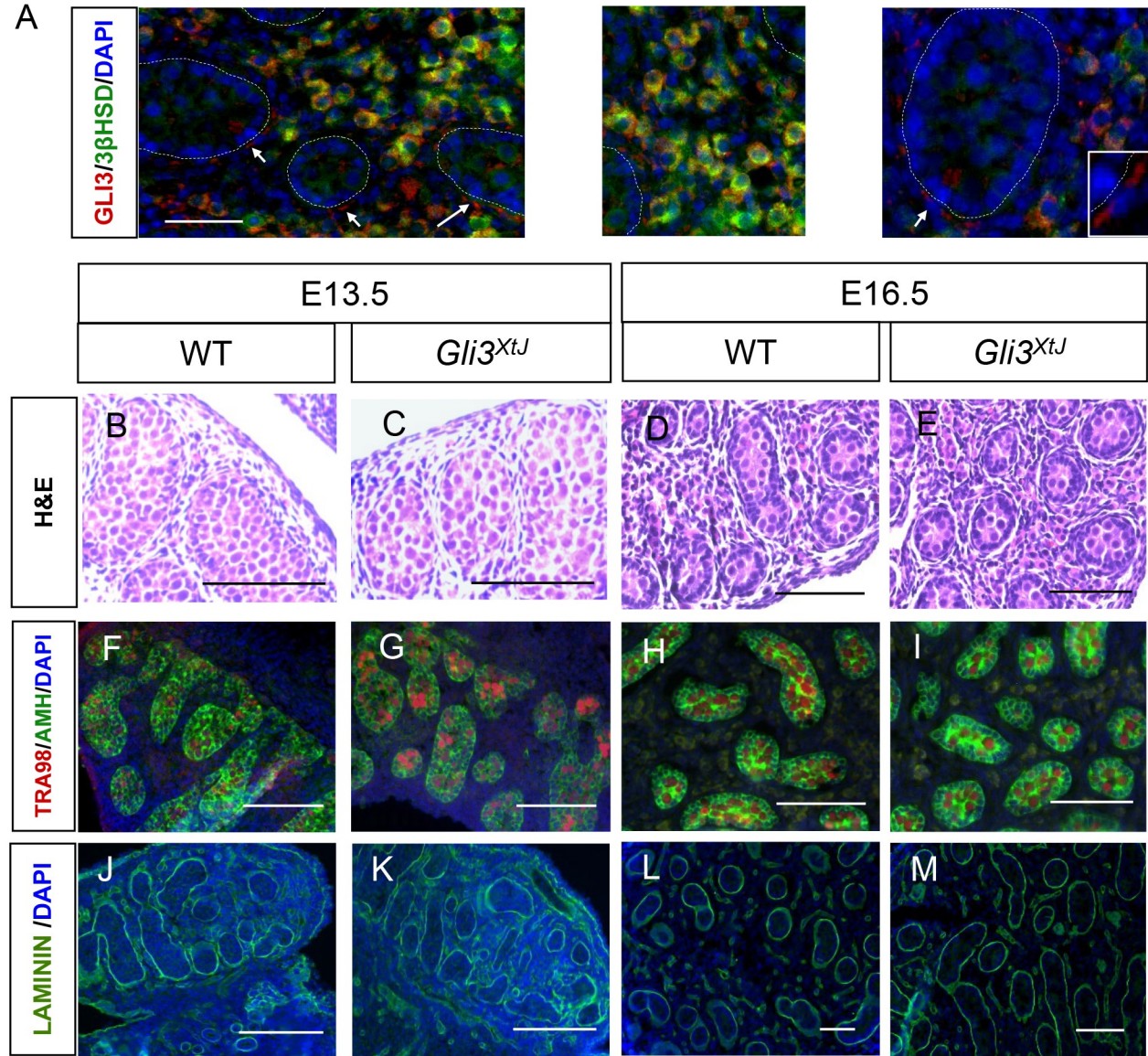

**Fig 4. *Gli3<sup>XtJ</sup>* mutant embryos have normal testicular morphology. A)** Double stain immunohistochemistry was used to detect β-galactosidase (red, marked GLI3) along with 3βHSD (Leydig cell marker, green) in E16.5 testes of *Gli3-LacZ* embryos. DAPI marks cell nuclei in blue. Yellow denotes co-localization in fetal Leydig cells. Arrows denote peritubular myoid cells; testis cords are outlined by dotted lines. **B-M)** Representative images of testis sections at E13.5 and E16.5 comparing wild type and *Gli3<sup>XtJ</sup>* mutant embryos—H&E staining (B-E), immunostaining for germ cell marker, TRA98 (red) and Sertoli cell marker, AMH (green) (DAPI nuclear counterstain, blue; F-I), and immunostaining for laminin (green) (DAPI nuclear counterstain, blue; J-M) to highlight intact basement membranes.

were not different between wild type and *Gli3<sup>XtJ</sup>* mutant testes at both E13.5 and E16.5 (S3D and S3E Fig). Altogether, these data suggest that deficient hormone production in *Gli3<sup>XtJ</sup>* mutant testes is caused by an insufficient number of steroidogenic cells as defined by expression of 3βHSD.

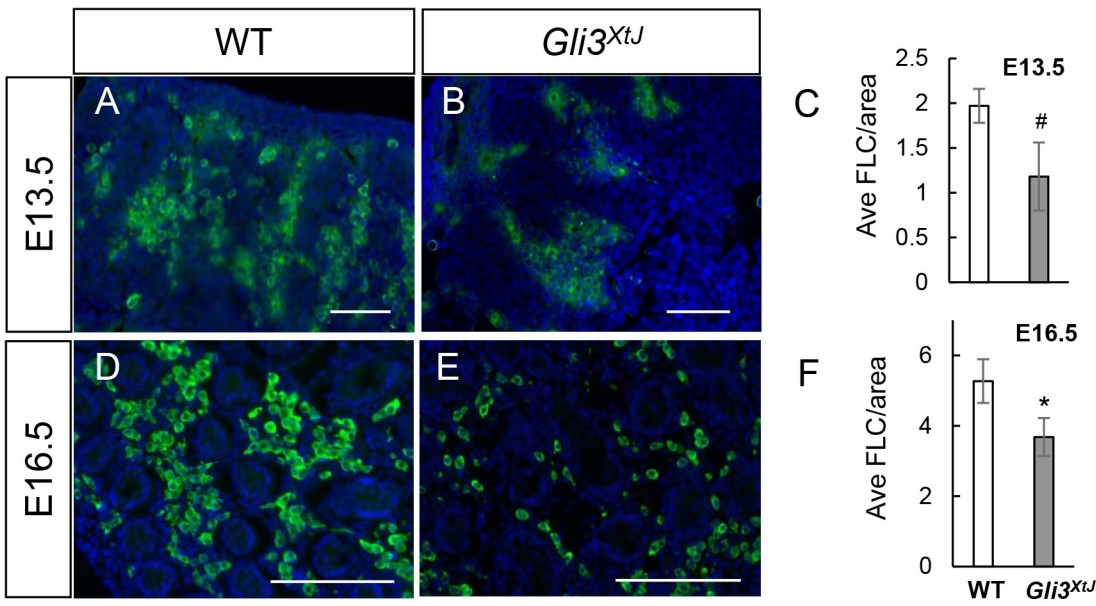

**Fig 5. *Gli3^{XtJ}* mutant embryos have fewer fetal Leydig cells.** Representative images of 3βHSD immunostained fetal Leydig cells (green) with DAPI nuclear counterstain (blue) at E13.5 and E16.5 in wild type (A,D) and *Gli3^{XtJ}* mutant (B,E) testes. **C, F)** Cells positive for 3βHSD were quantified from stained sections and results are represented as mean ± SEM. Biological replicates are as follows: E13.5 WT n = 5, *Gli3^{XtJ}* n = 5; E16.5 WT n = 5, *Gli3^{XtJ}* n = 3. Student's t-test, #: $p < 0.1$, *: $p < 0.05$. Scale bar: 100 μm.

## Fetal Leydig cells in *Gli3^{XtJ}* mutant embryos are dysfunctional

We next investigated deficits of the fetal Leydig cell population in the context of the developing testis. First, we evaluated whether the *Gli3^{XtJ}* mutation affected the genes within the hedgehog signaling pathway itself. Testicular transcripts of *Dhh* increased at E13.5 and remained unchanged at E16.5 (Fig 6A). *Gli1* and *Ptch1*, both downstream targets of hedgehog signaling, were significantly decreased by E16.5. In addition, there was a stepwise decrease in *Gli2* from E13.5 and E16.5 (Fig 6A). These results suggest that the loss of *Gli3* perturbs hedgehog signaling over time. One critical feature to hedgehog signaling is its dependency on the primary cilium for its function and GLI processing [39]; therefore, we evaluated features of primary cilia within wild type versus *Gli3^{XtJ}* mutant fetal testes. Results showed similar staining patterns of cilia among interstitial cells in general and on marked fetal Leydig cells (S4A Fig). Next, we investigated transcript profiles of *Foxj1* and *Rxp3*, genes that are involved in cilia biogenesis and responsive to hedgehog signals [40] and found no change in expression of either gene (S4B Fig). Collectively, these data suggest that ciliopathy is not the cause for disrupted hedgehog transcript profiles in *Gli3^{XtJ}* mutant testes.

*Gli3^{XtJ}* mutant testes produce less *Insl3* and testosterone indicating dysfunctional fetal Leydig cells; therefore, we performed qRT-PCR to examine production of genes related to steroid synthesis. In the fetal testis, Sertoli cell-derived DHH signals stimulate differentiation of SF1-positive progenitor cells into fetal Leydig cells, which is heralded by upregulation of *Sf1* and then SF1-dependent steroidogenic enzymes [8,41]. Our results show that transcript levels of *Sf1*, *Star*, *Cyp11a1* and *Cyp17a1* were significantly decreased in mutant testes at both E13.5 and E16.5 (Fig 6B). This result could be interpreted as a true decrease in gene expression or it could reflect a decrease in fetal Leydig cell number, or both. To adjust, transcript levels reported in Fig 6B were normalized to Leydig cell numbers as quantified by 3βHSD staining (Fig 5C and 5F). Raw values without normalization indicated that the transcript levels were

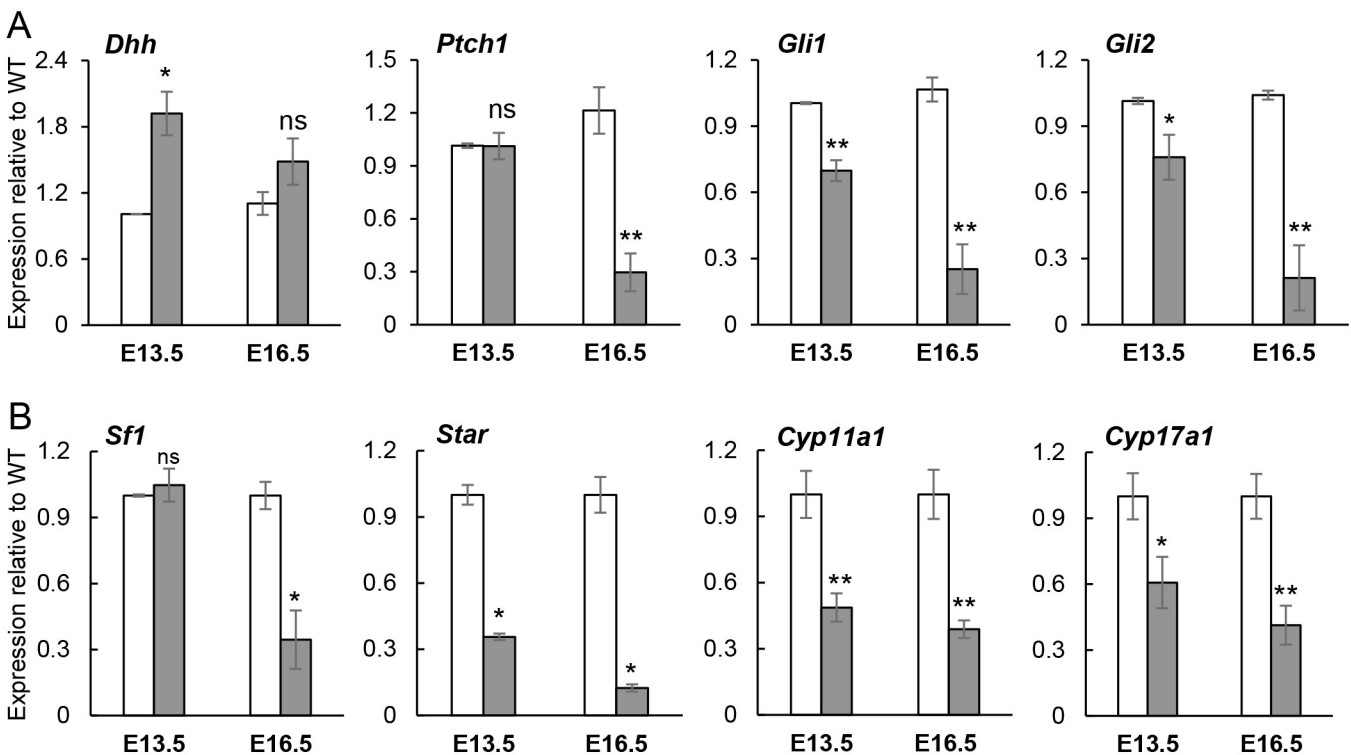

**Fig 6. Hedgehog and steroidogenic pathway genes are decreased in *Gli3^XtJ* testes.** RT-qPCR comparing wild type (white bars) and *Gli3^XtJ* mutant (gray bars) testis mRNA expression for hedgehog (**A**) and steroidogenic (**B**) pathway genes at E13.5 and E16.5 and expressed as fold change from wild type control at each time point and reported as the average ± SEM from n = 3–4 biological replicates. Student's t-test, *: p<0.05, **: p<0.01, ns: non-significant.

largely reflected by the decrease in cell numbers (S5 Fig); however, normalized data indicated an additional 25–40% decrease, especially at E16.5. Altogether, these data suggest that although fewer fetal Leydig cells can be identified in *Gli3^XtJ* mutant testes, their function is affected and worsens over time.

## Exogenous *Gli3* restores the fetal Leydig cell transcript profile in *Gli3^XtJ* mutant testes

SF1 is known as the master regulator of genes within the steroidogenic pathway [42,43]; of interest, both GLI2 and GLI3 have been reported to stimulate distinct steroidogenic genes to promote steroid synthesis [44]. To test whether replacement of GLI3 could restore expression of *Sf1* and other steroidogenic genes, we transfected *Gli3^XtJ* mutant testes using microinjection followed by electroporation [45] with CMV driven expression vectors encoding either EGFP (negative control), or Gli3FL, a constitutively active form of GLI3 [46]. Testes were harvested and transfected at E14.5 followed by 48 hours culture and then processed for RNA and copy number qPCR to quantify transcript numbers of each gene. Reintroduction of Gli3FL increased *Gli3* transcript levels from 0% to 40% of wild type control (Fig 7A). The remainder of transfection data was focused on results within *Gli3^XtJ* testes. As expected, transfection of Gli3FL plasmids increased expression of hedgehog-responsive gene, *Gli1*, and had no effect on the negative control, *Sox9*. In addition, Gli3FL expression had no impact on *Gli2* or *Insl3* (Fig 7B). Reintroduction of *Gli3* expression stimulated a significant increase for each fetal Leydig

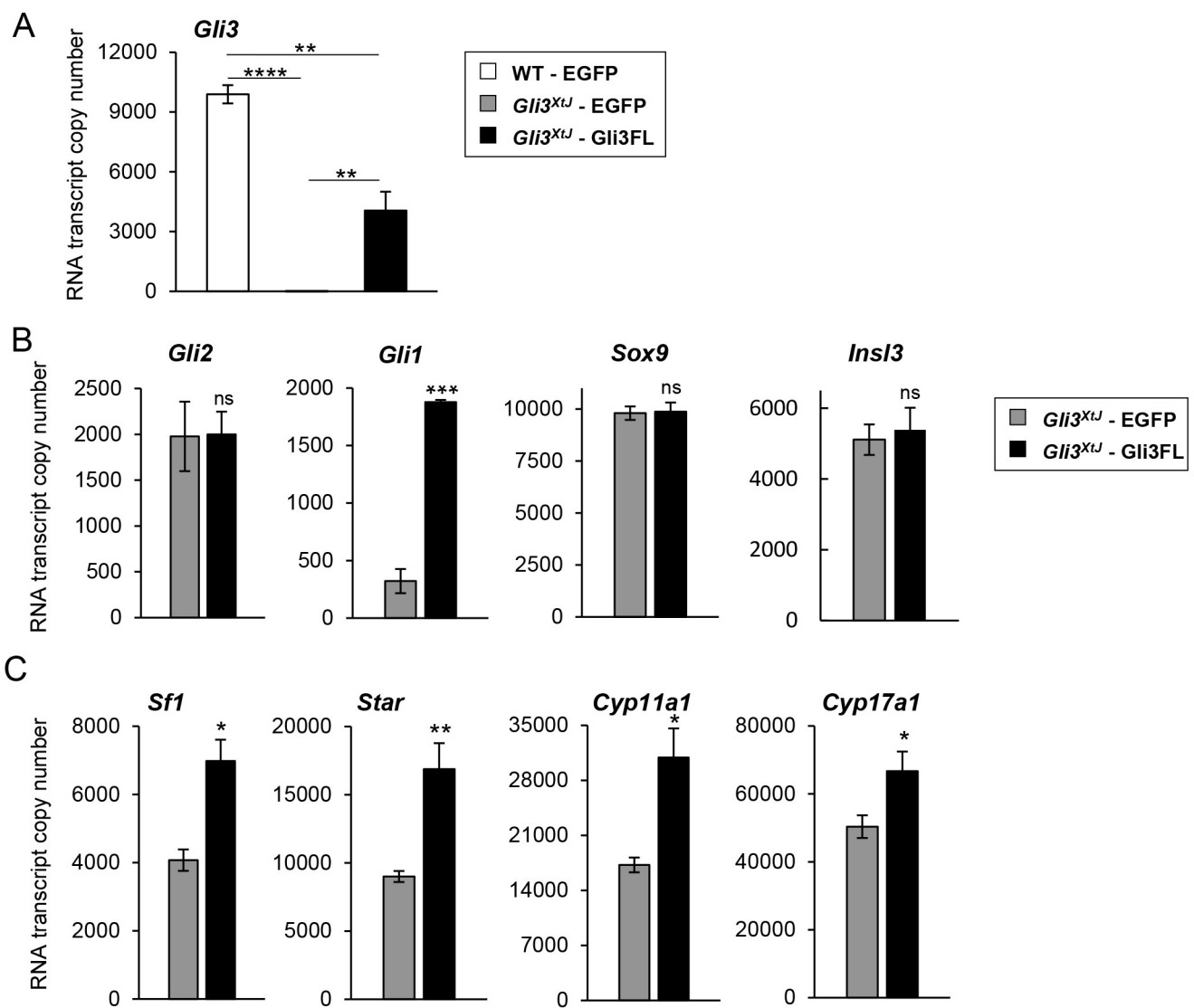

**Fig 7. Reintroduction of pGli3FL plasmid DNA restored expression of steroidogenic genes. A)** Wild type and *Gli3^XtJ* mutant testes were transiently transfected with pEGFP-C2 or pEGFP-Gli3FL plasmids and transfection efficiency was analyzed by quantifying the transcript numbers of *Gli3* using copy number qPCR. Results are represented as mean ± SEM from n = 3–4 biological replicates. One-way ANOVA followed by post-hoc Tukey's test, **: p<0.01, ****: p<0.0001. **B,C)** *Gli3^XtJ* mutant testes were transiently transfected with pEGFP-N1 (gray bars) or pEGFP-Gli3FL (black bars) plasmids. Transcript numbers of *Gli2* and *Sox9* (negative controls), *Gli1* (positive control), *Insl3* (**B**), and steroidogenic genes (**C**) were measured in the transfected testes. Results are represented as mean ± SEM from n = 3–4 biological replicates. Student's t-test, *: p<0.05, **: p<0.01.

cell gene associated with steroidogenesis (Fig 7C). Based on these data, we conclude that hormone dependent male sex differentiation requires activated GLI3 to promote fetal Leydig cell differentiation, maintenance, and testosterone synthesis.

## Discussion

Results from this study showed that *Gli3^XtJ* mutant male embryos exhibit several characteristics associated with differences in male sex differentiation including decreased anogenital distance, delayed testicular descent, and urethral tube defects, including hypospadias. Differences

in sex development, especially those related to male sex differentiation, have significantly increased in incidence over the last decade and few explanations have emerged [47]. Each of these sex characteristics requires activated androgen receptor activity along with other signaling pathways, including hedgehog, within critical developmental windows. For example, independent studies have shown that perturbation of androgen receptor activity or locally acting signals, such as hedgehog and wnt, within defined developmental stages, can cause disrupted genital tubercle and phallus growth [21,48]. These and other studies illustrate the interconnectedness of androgen activity and other pathways in male sex differentiation [49]. While there are some clear examples related to specific genetic defects, the most likely explanation includes complex interactions between genetic defects and environmental factors, which together, also impact epigenetic codes. For example, only 5–29% of GCPS and PHS patients exhibit male disorders of sex differentiation (Genetic and Rare Disease Information Center, US Department of Health & Human Services). While these mutations have been described as completely penetrant with variable severity, our data suggest that variation in presentation of these specific phenotypes might also be explained by a sensitivity of the *GLI3* mutation in certain patients to another layer of vulnerability imposed by androgen action.

Each of the disorders identified in $Gli3^{XtJ}$ mutant male embryos have been associated with both androgen and hedgehog signaling deficiencies [25,50–52]; therefore, we hypothesized that the hedgehog and androgen pathways work together to promote male sex differentiation. Our results confirm this hypothesis, but also show that the pathway interactions are unique to each tissue in question. We modeled androgen and hedgehog contributions to fetal Leydig cell differentiation, testis descent, prostate and external genitalia formation (Fig 8). Besides testis descent, which we found to be independent of hedgehog signaling, all other male sex differentiation requires the androgen-activated androgen receptor (AR) along with GLI-mediated hedgehog (HH; Sonic, Desert, or Indian) activity. The fetal Leydig cell requires DHH to stimulate androgen synthesis and GLI3A must be present for optimal productivity. These circulating androgens will then bind AR to stimulate IHH to promote male genital tubercle development and cooperate with SHH in prostate bud formation. Timing of these events is crucial.

External genitalia and AGD defects within $Gli3^{XtJ}$ mutant male embryos varied in severity, ranging from indistinguishable from wild type to severe penoscrotal hypospadias. In addition, prenatal administration of DHT did not rescue external genitalia defects in $Gli3^{XtJ}$ mutants. One explanation for this outcome could be insufficient delivery of DHT to embryos; however, this is unlikely because we detected retained Wolffian duct structures in female embryos from litters of each treated dam, validating excess androgen exposure (S6 Fig). Previous studies have established that androgens promote male genital tubercle differentiation with the critical prenatal window defined as E13.5 –E16.5 in mice [21]. These studies were achieved with conditional deletion of the *Ar* gene and with the use of an androgen receptor antagonist. Sex steroid receptors are known to stimulate both Sonic and Indian hedgehog in numerous cell types, including those in the developing genital tubercle. Further, evidence that the hedgehog pathway is downstream of the androgen receptor is supported by findings that *Gli2* and *Gli3* mutant mouse models, in addition to temporally controlled deletion of *Shh* between E13.5 and E15.5, cause hypospadias and a urethral phenotype that is strikingly similar to the hypospadias observed in $Gli3^{XtJ}$ mutants [10,11,52]. It is currently not known whether exogenous androgens could alleviate or bypass these hedgehog-mediated defects. Although we did not evaluate transcripts of other factors downstream of hedgehog, our finding that external genital defects in the $Gli3^{XtJ}$ mutants could not be rescued by exogenous DHT suggests that this would not be the case. Nonetheless, because our animals did have some circulating androgens, albeit very low, we acknowledge the possibility that low levels of androgens could be sufficient to promote expression of hedgehog pathway genes in the face of the $Gli3^{XtJ}$ mutation. Supplementation

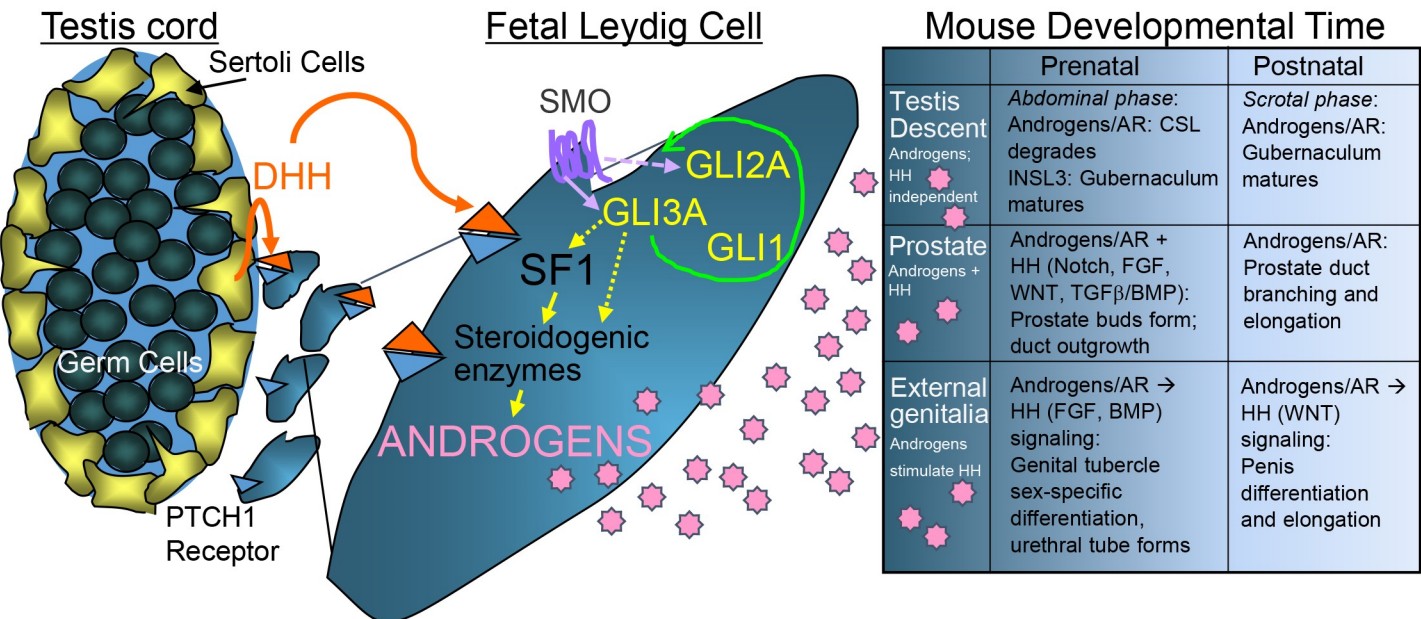

**Fig 8. GLI3A and androgens cooperate to promote male reproductive tract maturation.** Sertoli cells produce DHH, which diffuses to the testis interstitium where it interacts with PTCH1 receptors to stimulate fetal Leydig cell differentiation. Bound PTCH1 releases SMO to stabilize the activated GLI3 transcription factor (GLI3A), likely along with GLI2A, which indirectly (dashed arrow) stimulates *Sf1 (Nr5a1)* and other steroidogenic genes to initiate androgen production. Fetal Leydig cells produce androstenedione, which is converted to testosterone in Sertoli cells. GLI3A is necessary to maintain fetal Leydig cell identity by promoting the fidelity of both the hedgehog and steroidogenic enzyme pathways (green loop arrow). Testosterone is released into fetal circulation to promote testis descent, prostate and external genitalia formation (detailed descriptions in Box). Each downstream developmental event needs discrete levels of testosterone at specific times. While both abdominal and scrotal phases of testis descent are independent of HH signaling, prostate and external genitalia development both require HH. Prenatal prostate buds can form with minimal androgen exposure and without HH; although, elevated levels of each are likely important for later developmental stages. Male genital tubercle development is initiated by activated AR, which in turn, stimulates an increase in production of HH and other signaling factors. *Gli3^{XtJ}* mutant embryos are embryonic lethal precluding analysis of postnatal development; however, our analysis of male sex characteristics during prenatal development highlight cooperation and interactions between androgen/AR and HH signaling pathways.

with DHT however, failed to alleviate those with the phenotype indicating that the *Gli3^{XtJ}* mutation is sufficient to impede proper development. Thus, we conclude that the variability in urogenital and AGD defects is caused by the unpredictable circulating concentrations of androgens layered upon the *Gli3^{XtJ}* mutation. Taken together with the inability of DHT to rescue the hypospadias in *Gli3^{XtJ}* mutants, our data support the hypothesis that the urethral defect is a direct result of disrupted hedgehog signaling in the genital tubercle.

The impact of varying concentrations of circulating androgens is further supported by our observation that prostate bud emergence was not different between *Gli3^{XtJ}* mutant and wild type male embryos, suggesting sufficient androgen exposure specific to prostate bud development by E16.5. Prostate bud formation is complete by E18.5 and is sensitive to low levels of androgens, but independent of *Gli3* [53–56]. Mouse urogenital morphogenesis has been separated into pre- and postnatal phases with distinct sensitivity to both androgens and estrogens [19,21]. Urethral tube defects in particular are exquisitely sensitive to androgen/AR activity during prenatal development within a critical window between E13.5–16.5 [21]. Future experiments designed to correlate testicular testosterone concentrations to specific gradations of defects within individual *Gli3^{XtJ}* mutant male embryos would be helpful to validate these conclusions.

Testicular descent is a critical event in male sexual differentiation and like urogenital morphogenesis, occurs in two distinct stages separated by the time of birth in mice [57]. The

inguinoscrotal phase of the testicular migration occurs after birth in mice [22] and hence cannot be evaluated in *Gli3^{XtJ}* mutants. *Gli3^{XtJ}* mutant male embryos exhibited a failure in the transabdominal phase of descent by E16.5. Prenatal descent relies on a coordinated regression of the CSL, which is degraded in response to androgens, and the pull of the gubernaculum as it differentiates in response to INSL3 [23,27]. The sole source for both INSL3 and androgens is the fetal testis, and both were significantly diminished in *Gli3^{XtJ}* mutant testes. We did not detect structural defects within the CSL or gubernaculum or find evidence that the ligaments were dependent on the hedgehog pathway for morphogenesis at early stages of embryonic development. We observed that exogenous DHT exposure had a significant impact on the integrity of collagen content within the androgen-dependent CSL although enhanced collagen degradation was insufficient to promote timely descent. The essential role of androgens in the regression of the CSL comes from animal studies that show that disturbances in androgen synthesis or action result in retained CSL [23,58]. Although it is reported that the CSL is a fibromuscular structure composed of smooth muscle fibers and abundant collagen [59], there are few details regarding the collagen content of the CSL. To our knowledge, very few studies have administered exogenous DHT to rescue failure of CSL regression [60]; our studies show that DHT was able to reduce the collagen content of the CSL suggesting an AR-mediated program in collagen degradation. Testis descent was observed in most *Gli3^{XtJ}* mutant male embryos by E17.5; therefore, it is likely that co-supplementation of DHT and INSL3 would be required to fully rescue the defective phenotype. Based on these results, we conclude that the hedgehog and androgen pathways do not converge directly within the ligamentous structures that mediate testis descent and that this process requires relatively higher concentrations of androgens along with sufficient INSL3 for an optimal outcome. Thus, the failure in the transabdominal phase of testis descent in *Gli3^{XtJ}* mutant male embryos suggests that the hedgehog/androgen interface must occur at the level of the fetal testis.

It has long been established that DHH mediated Smoothened activity stimulates fetal Leydig cell differentiation [8,61]. Further, inhibition of all three GLI transcription factors downstream of Smoothened causes impaired fetal Leydig cell differentiation, but individual elimination of *Gli1* or *Gli2* had no obvious impact [12]. These data prompted our investigation into the impact of *Gli3* on fetal testis development. Given the impact of the *Gli3^{XtJ}* mutation on fetal Leydig cell hormone production, we were not surprised to find fewer cells positive for the steroidogenic enzyme, 3βHSD, or decreased expression of *Insl3*, *Sf1*, and other steroidogenic enzyme genes. We have identified putative GLI binding sites in enhancer regions in loci of *Sf1* including the putative fetal Leydig cell enhancer reported by Shima *et al.* [62]; however, transfection analysis in fetal testes (as in Fig 7) and other Leydig cell lines have not supported their specificity. Our data suggest that GLI3 contributes to stabilization of the fetal Leydig cell identity likely through indirect means (Fig 8). At least the cells that were present produced enough INSL3 and androgens to allow prostate bud formation, eventual testis descent, and some success in external genitalia formation. We were, however, surprised to discover evidence to suggest that the hedgehog pathway was significantly impaired. At E13.5, transcripts for Sertoli cell-derived *Dhh* were increased 2-fold in an apparent response to deficient downstream effects on hedgehog responsive cells as indicated by diminishing *Ptch1* and *Gli1* target gene transcripts. In addition, *Gli2* expression decreased significantly over time, but we did not detect evidence for ciliopathy. These results, taken together with the fading expression of hormone-related genes, suggested that the fetal Leydig cell population was losing its identity over time.

Fetal testes were evaluated by immunofluorescence and qPCR for evidence that would suggest a shift from the fetal Leydig cell identity to a progenitor or other cell type. We did not observe any changes that would indicate accumulation of progenitor cells or defects in other

cell types that are associated with fetal Leydig cell differentiation. Other studies with altered hedgehog/GLI signaling activity have shown evidence for target cell dedifferentiation. We found that reintroduction of a plasmid encoding the GLI3 active form into the *Gli3^XtJ* mutant testis had no impact on *Insl3*, but significantly increased transcripts required for androgen production. This result was reflected in the functional output of fetal Leydig cells over time. Although both *Insl3* and testosterone were substantially decreased at E13.5 and E16.5 in *Gli3^XtJ* mutant compared to wild type testes, *Insl3* levels increased at the same rate over time in testes of both genotypes. In contrast, testosterone production remained stagnant in mutant compared to wild type testes, suggesting fundamental differences in regulation of each hormone. Notably, the addition of GLI3A did not impact expression of *Gli2*; however, *Gli1* increased as expected. Results suggest that appropriate expression of GLI3A is critical for maintenance of hedgehog signaling capacity and fetal Leydig cell identity. Thus, the fetal Leydig cell within the developing testis represents the foundation for hedgehog/androgen interactions in mediating male sex differentiation.

In summary, our detailed analysis of the differences in male sex differentiation caused by the *Gli3^XtJ* mutation highlights the unique requirements for the interface between hedgehog and androgen action for each tissue. Ultimately, the fetal testis must initiate the program. At a critical time point in fetal testis development, paracrine DHH signals initiate Smoothened activity and hence, GLI3A, likely in concert with GLI1 and GLI2, to stimulate transcription of *Insl3*, *Sf1* and the steroidogenic enzyme pathway genes that provide the foundation for INSL3 and androgen synthesis within fetal Leydig cells. GLI3A is required to maintain this activity. Ultimately, these hormones must reach specific threshold levels for each tissue and in some cases, such as external genitalia and the anogenital region, testis-derived androgens and local hedgehog, among other signals, must coordinate with exquisite timing for appropriate male reproductive tract development.

## Materials and methods

### Animals

*Gli3^XtJ* mice were originally obtained from Jackson Laboratories and have been maintained at the University of Wisconsin for >10 years by breeding into the C57BL/6J background. Genotyping for the wild type and mutant alleles was performed by PCR as published [15,63]. Embryonic testes were differentiated from ovaries by visualization of the characteristic coelomic vessel in the testis. Heterozygous mice were bred for timed pregnancies, and the presence of a vaginal plug the next morning was considered 0.5 day of gestation (E0.5). Embryos were harvested at the stated days and gonads were processed as described. Embryonic males were distinguished by the presence of classic testicular structures including testis cords and the coelomic vessel. In the event that tissues were processed without gonad visualization, tail snips were collected to harvest genomic DNA and perform PCR for *Sry* (see S1 Table for primers) and the *Gli3^XtJ* mutation. All the procedures described were reviewed and approved by the Institutional Animal Care and Use Committee at the University of Wisconsin and were performed in accordance with the NIH Guiding Principles for the Care and Use of Laboratory Animals.

### Morphological measurements

The urogenital system with the gonad and the external genitalia were examined at the indicated time points with a Leica M165FC stereo microscope and images were taken with a Leica DFC295 camera (Leica Microsystems) to enable gross morphological evaluation. Images were coded and anogenital distance was measured by a different investigator that was blinded to

genotype from the image. Anogenital distance (AGD) was quantified by measuring the length from the caudal base of the genital tubercle to the anterior aspect of the anus. Wild type and *Gli3*$^{XtJ}$ male fetuses were analyzed on E16.5.

## External Genitalia

Embryos were fixed overnight in 4% paraformaldehyde and dissected to remove limbs and tail in order to provide unobstructed views of anogenital region. Topographic anatomy was examined under a Leica MZFLIII stereo dissecting microscope and images were recorded using a ProgRes C14 Plus CCD camera (Jenoptik). Specimens were then contrast stained in 2.5% aqueous Lugol's Iodine (IKI) for high resolution x-ray nanoscale computed tomography (nanoCT). Scanning was performed at the University of Florida's Nanoscale Research Facility on a Phoenix V|tome|X M scanner (General Electric) with a 180 kv nanofocus x-ray tube. Radiographs were imported into Phoenix DatosOS|X reconstruction software to produce tomograms, and CT datasets were analyzed, and images produced using VGStudioMax 3.3 software (VolumeGraphics).

## *In situ* hybridization (ISH)

Samples were fixed in 4% paraformaldehyde overnight at 4°C and whole-mount *in situ* hybridization was performed according to standard procedures [64]. ISH and genotyping were conducted by separate labs to ensure that phenotypic analysis was blind to genotypes. *Nkx3-1* riboprobes were generated as described previously [64]. The digoxigenin-labeled probe was detected by using an alkaline phosphatase conjugated anti-digoxigenin antibody.

## Histology

Testes and embryos were fixed in 4% paraformaldehyde overnight at 4°C. Tissues were either embedded in paraffin or in OCT freezing media (Tissue-Tek). Paraffin sections were stained with hematoxylin and eosin (H&E) for histological analysis. Sections from at least 4 animals from different litters for each genotype were examined, representative images are shown.

For immunohistochemical analysis, paraffin or frozen sections were stained with appropriate antibodies (S2 Table) according to standard procedures. Briefly, sections were incubated at 4°C overnight with primary antibodies followed by incubation at room temperature with secondary antibodies for 1 hr. Slides were counterstained with DAPI and mounted on glass coverslips.

For collagen staining, paraffin sections were stained with picro-sirius red and mounted with Richard-Allan mounting media as described previously [30]. All images were captured on a Leica SP8 confocal microscope or a Keyence BZ-X700 microscope and processed with ImageJ or Adobe Photoshop. Collagen fiber metrics were measured using CT-FIRE fiber detection software (LOCI; Madison, WI) [31].

## ELISA

Testes were dissociated with Tissue Protein Extraction Reagent (Pierce) using a rotary homogenizer. Testosterone ELISA was performed as previously published [65]. Briefly, Immunosorb 96-well plates (Nunc) were coated with goat anti-mouse immunoglobulin followed by a mouse antibody against testosterone (Biostride). Sample lysates were then added to plates and allowed to bind for 1.5 hr after which HRP-conjugated testosterone was added and allowed to compete for antibody binding. Following a wash, 3,3′,5,5′-tetramethylbenzidine was added to develop

color and quenched with 0.5M sulfuric acid. HRP-conjugated testosterone was a kind gift of Dr. Milo Wiltbank, University of Wisconsin-Madison.

## RT-qPCR and Copy number qPCR

Total RNA was extracted using the RNeasy mini kit (QIAGEN) and 500ng RNA was used for cDNA synthesis by SuperScript II cDNA First-Strand Synthesis System (Invitrogen) using random primers. The cDNA was amplified by real-time quantitative PCR using SYBR Green (Bio-Rad) as fluorophore in a CFX96 real-time thermal cycler (Bio-Rad). A 10 µl qRT-PCR reaction volume was used with 2 µl of 1:4 diluted cDNA and 300 nM primer. Results were calculated using comparative $C_T$ method and normalized to *36B4* (*Rplpo*) from at least three separate biological replicates. All reactions were performed in duplicates. Primer sequences can be found in S1 Table.

Copy number qPCR was performed to determine the absolute copy number of target gene by relating the $C_T$ value to a standard curve. The composite gBlock (IDT Technologies) is used as the quantification standard, which includes all the target gene sequences (PCR products) synthesized on the same genomic block (S3 Table) [66]. Total RNA was extracted using the RNeasy mini kit (QIAGEN) and 500ng RNA was used for multi-primed cDNA synthesis with a cocktail mix of 1 µM each of gene specific reverse primer. A standard curve was generated from tenfold serial dilution series of gBlock fragments ranging from $10^2$ copies/µL to $10^6$ copies/µL. The exact copy number of the unknown samples was calculated from the standard curve using the formula: Absolute transcript copy number = $10^{\wedge}[(C_T-(\text{y-intercept}))/\text{slope}]$.

## 5α-dihydrotestosterone treatment

Pregnant dams were treated daily with sesame oil or DHT (1 mg/kg body weight), starting on E12.5 and the embryos were examined at E16.5. Appropriate embryonic absorption of DHT was validated by examining female embryos for the presence of a retained Wolffian duct. The Wolffian duct is precursor of the male-associated reproductive tract and its maintenance is dependent on the presence of androgens. Female embryos from DHT treated dams were harvested and their reproductive tract was examined under phase contrast microscopy for the presence of both the Müllerian and Wolffian ducts (S6 Fig).

## Fetal testis transfection analysis

Fetal testis transfection assays were performed as previously described [45]. Testes were harvested from embryos at E14.5 and transfected with pEGFP-C2 (Clontech) or pEGFP-Gli3FL constructs. Gli3FL plasmid was a kind gift from Dr. Leslie Biesecker, NIH. Approximately 0.2 µl of the plasmids (4 µg/µl) was injected into the gonad at three different sites. An aliquot of 20 µl of sterile PBS was placed on the testes for electroporation. Immediately thereafter, five square electrical pulses of 65 V, 50 msec each at 100-msec intervals, were delivered through platinum electrodes from an electroporator (BTX Electro Square Porator, ECM830). After electroporation, testes were placed into 500 µl of Dulbecco minimal Eagle medium (DMEM) supplemented with 10% FBS and 1% Penicillin-Streptomycin, and cultured at 37°C with 5% $CO_2$/95% air for 48 hr and then rinsed with PBS and processed for copy number qPCR analysis.

## Statistical analyses

Statistical differences were assessed using a two-tailed Student's t-test assuming unequal variances. Results were considered statistically different if p-values were $\leq 0.05$. Results of $p < 0.1$ are also reported. One-way ANOVA Post-hoc Tukey was performed where appropriate.

## Supporting information

**S1 Fig. *Prenatal DHT administration did not rescue the external genitalia phenotype in Gli3^XtJ^ mutant male embryos.*** External genitalia of *Gli3^XtJ^* males harvested from pregnant dams that were treated daily with sesame oil (A) (n = 7) or DHT (1 mg/kg body weight) (B) (n = 8) beginning at E12.5. Arrows indicate ectopic opening of hypospadiac urethra. Embryos were harvested at E16.5 or E17.5 and imaged using whole-mount light microscopy.
(TIF)

**S2 Fig. Gubernaculum development is independent of hedgehog signaling; *Rxfp2* expression in the gubernaculum and androgen receptor expression in the CSL were unaffected in *Gli3^XtJ^* mutants. A)** β-galactosidase (blue) expression in urogenital tracts of a *Gli1-LacZ* P0 male. The white arrow highlights positive LacZ stain in the vas deferens and the black arrow points out the lack of staining in the gubernaculum **B)** Expression of INSL3 receptor, *Rxfp2*, is unaffected in *Gli3^XtJ^* gubernaculum at E16.5 compared to the wild type. Results are normalized to wild type control and are represented as mean ± SEM from three biological replicates. Student's t-test, ns: non-significant. **C)** IHC with anti-androgen receptor (AR) antibody shows similar expression pattern in wild type and *Gli3^XtJ^* mutant embryos. Arrows denote cranial suspensory ligaments (CSL); epididymis (E), intestine (Int), kidney (K), testis (T).
(TIF)

**S3 Fig. *Gli3^XtJ^*mutation does not affect survival of fetal Leydig cells or fate of progenitors. A)** Expression of cleaved caspase 3 with DAPI nuclear counterstain at E16.5 in wild type and *Gli3^XtJ^* testes. P0 ovary is included as a positive control, arrows highlight positive staining for cleaved caspase 3. **B,C)** Expression levels of Wt1 and Nr2f2 were analyzed by IHC (B, Scale bar: 100 μm) and RT-qPCR (C). **D, E)** Transcript levels of Notch signaling pathway genes (D) and PDGF signaling pathway genes (E) were analyzed by RT-qPCR. Results are normalized to wild type controls at each age and are represented as mean ± SEM from three biological replicates. Student's t-test, #: p<0.1, ns: non-significant.
(TIF)

**S4 Fig. *Gli3^XtJ^* mutant testes have normal cilia morphology. A)** Representative images of acetylated γ-tubulin (green) and 3βHSD immunostained (red, counterstained with DAPI) wild type and *Gli3^XtJ^* testes at E16.5 showing similar appearance of cilia. Scale bar: 100 μm. **B)** Transcript levels of genes involved in ciliary biogenesis, *Foxj1* and *Rxp3* are unaffected in *Gli3^XtJ^* testes at E13.5 and E16.5 compared to wild type. Results are reported as compared to wild type controls at each age and represented as mean ± SEM from three biological replicates. Student's t-test, ns: non-significant.
(TIF)

**S5 Fig. Expression of steroidogenic genes not normalized to fetal Leydig cell numbers.** RT-qPCR comparing wild type and *Gli3^XtJ^* testes mRNA expression of steroidogenic genes at E13.5 and E16.5 normalized to *36b4* (*Rplpo*) and expressed as fold change from wild type testes ± SEM, from n = 3–4 biological replicates. Student's t-test, **: p<0.01, ***: p<0.001.
(TIF)

**S6 Fig. Prenatal DHT administration results in retained Wolffian ducts in *Gli3^XtJ^* mutant female embryos. A,B)** Pregnant dams (n = 5) were treated daily with sesame oil or DHT (1 mg/kg body weight), starting on E12.5 and harvested at E16.5. Female reproductive tracts were examined under phase contrast microscopy. Arrows indicate epididymis head (A) and the presence of both the Müllerian and Wolffian ducts (B).
(TIF)

**S1 Table. Primer list.**
(DOCX)

**S2 Table. Antibody list.**
(DOCX)

**S3 Table. Gblock sequence and primer list.**
(DOCX)

**S1 Methods. Description of methods used for supplemental figures.**
(DOCX)

## Acknowledgments

Kyle Wegner for his help with picrosirius red staining and CT-FIRE software. Jorgensen Lab Members and the Developmental Endocrinology Research Group, especially Rob Lipinski, for critical commentary and support. Nicole Franks and Dr. Benjamin Allen (University of Michigan) for *Gli3-LacZ* gonads.

## Author Contributions

**Conceptualization:** Anbarasi Kothandapani, Samantha R. Lewis, Alexander I. Agoulnik, Martin J. Cohn, Joan S. Jorgensen.

**Data curation:** Anbarasi Kothandapani, Samantha R. Lewis, Jessica L. Noel, Abbey Zacharski, Kyle Krellwitz, Anna Baines, Stephanie Winske, Chad M. Vezina, Elena M. Kaftanovskaya, Alexander I. Agoulnik, Emily M. Merton, Martin J. Cohn, Joan S. Jorgensen.

**Formal analysis:** Anbarasi Kothandapani, Samantha R. Lewis, Jessica L. Noel, Anna Baines, Stephanie Winske, Chad M. Vezina, Elena M. Kaftanovskaya, Alexander I. Agoulnik, Martin J. Cohn, Joan S. Jorgensen.

**Funding acquisition:** Chad M. Vezina, Alexander I. Agoulnik, Martin J. Cohn, Joan S. Jorgensen.

**Investigation:** Anbarasi Kothandapani, Samantha R. Lewis, Jessica L. Noel, Abbey Zacharski, Kyle Krellwitz, Anna Baines, Stephanie Winske, Chad M. Vezina, Elena M. Kaftanovskaya, Alexander I. Agoulnik, Emily M. Merton, Martin J. Cohn, Joan S. Jorgensen.

**Methodology:** Anbarasi Kothandapani, Samantha R. Lewis, Jessica L. Noel, Abbey Zacharski, Kyle Krellwitz, Anna Baines, Stephanie Winske, Chad M. Vezina, Elena M. Kaftanovskaya, Alexander I. Agoulnik, Emily M. Merton, Martin J. Cohn, Joan S. Jorgensen.

**Project administration:** Alexander I. Agoulnik, Martin J. Cohn, Joan S. Jorgensen.

**Resources:** Chad M. Vezina, Alexander I. Agoulnik, Martin J. Cohn, Joan S. Jorgensen.

**Supervision:** Anbarasi Kothandapani, Samantha R. Lewis, Jessica L. Noel, Alexander I. Agoulnik, Martin J. Cohn, Joan S. Jorgensen.

**Validation:** Anbarasi Kothandapani, Samantha R. Lewis, Jessica L. Noel, Alexander I. Agoulnik, Martin J. Cohn, Joan S. Jorgensen.

**Visualization:** Anbarasi Kothandapani, Samantha R. Lewis, Jessica L. Noel, Kyle Krellwitz, Anna Baines, Stephanie Winske, Chad M. Vezina, Elena M. Kaftanovskaya, Alexander I. Agoulnik, Emily M. Merton, Martin J. Cohn, Joan S. Jorgensen.

**Writing – original draft:** Anbarasi Kothandapani, Samantha R. Lewis, Jessica L. Noel, Alexander I. Agoulnik, Martin J. Cohn, Joan S. Jorgensen.

**Writing – review & editing:** Anbarasi Kothandapani, Samantha R. Lewis, Jessica L. Noel, Anna Baines, Stephanie Winske, Chad M. Vezina, Alexander I. Agoulnik, Martin J. Cohn, Joan S. Jorgensen.

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
