## [Decision Letter · Decision Letter 0]

23 Feb 2020

Dear Dr Jorgensen,

Thank you very much for submitting your Research Article entitled 'GLI3 Resides at the Intersection of Hedgehog and Androgen Action to Promote Male Sex Differentiation' to PLOS Genetics. Your manuscript was fully evaluated at the editorial level and by independent peer reviewers. The reviewers appreciated the attention to an important topic but identified some aspects of the manuscript that should be improved.

We therefore ask you to modify the manuscript according to the review recommendations before we can consider your manuscript for acceptance. Your revisions should address the specific points made by each reviewer.

[LINK]

Yours sincerely,

Moira K. O'Bryan

Associate Editor

PLOS Genetics

Scott Williams

Section Editor: Natural Variation

PLOS Genetics

This manuscript has been reviewed by three researchers who work specifically in the area of sex determination. All agree this manuscript will make a valuable contribution to the field, as do I. The requested edits from reviewer 1 and 2 should be easily achieved. Some of reviewer 3's comments are more substantive, but at this point, I believe require discussion rather than additional experiments. The point about qualifying the conclusions is reasonable.

Reviewer's Responses to Questions

**Comments to the Authors:**

Reviewer #1: This manuscript explores how Hedgehog-GLI signalling and androgens interact to drive male urogenital development. In a series of carefully designed functional experiments the authors reveal the crucial role of GLI3 by using Gli3XtJ mutant mice. These mice have a range of phenotypes that affect the male reproductive tract including cryptorchidism and hypospadias. These mice showed lower levels of INSL3 (associated with testis decent) and testosterone. Testis development was otherwise normal in these mice but genes associated with androgen and INSL3 and the DHH pathway were disrupted. This suggests a failure of fetal Leydig cell differentiation and maintenance.

The study clearly shows that these mice have a reduction in the number of fetal Leydig cells which are the source of these testicular hormones and that their ability to function diminished over time. In an effort to separate out the effects of hormones and GLI3 they supplemented these mice with androgens. They show that this partially rescued testis decent but not the hypospadias. However, by adding a GLI3 activator into Gli3XtJ mutant mice testes they restored expression of the Hedgehog pathway and steroidogenic genes. This suggests that fetal Leydig cell function requires activated GLI3 for expression of genes critical to INSL3 and androgen synthesis.

My only real quibble is with data presented in Fig 6B showing a significant decrease in transcript levels of Sf1, Star, CYP11a1 and Cyp17a1. The authors suggest that this may be a real decrease in gene expression or due to fewer fetal Leydig cells or both. If expression levels were adjusted for Leydig cell number then raw values (without adjustment for cell number) showed a more severe impact on expression. The authors should conclude that this therefore means the reduction in expression is largely due to fewer Leydig cells. If this is correct it should be stated as such.

The overall message from this study is that appropriate expression of GLI3 is critical for the maintenance of hedgehog signalling and fetal Leydig cell identity. This allows us to build a new paradigm to understand male development. In the developing fetal testis, DHH signals likely initiate Smoothened activity and GLI3 which, in turn, stimulates Insl3, Sf1 and steroid pathway genes which lead to the synthesis of INSL3 and androgens within fetal Leydig cells. This ultimately leads to appropriate male urogenital development.

This is a well written and carefully conducted study that brings new insights to our understanding of male reproductive development. The rising incidence of male urogenital tract defects is discussed briefly. However, in the light of these findings I would like to see a more slightly more detailed discussion of how endocrine disruptors and hedgehog morphogens may impact on male urogenital development.

Reviewer #2: The paper by Kothandapani et al describes the male reproductive phenotype of GLI3 mutant mice. Since the phenotypes were consistent with a loss of androgen production in this model they supplemented back DHT and observed that some, but not all phenotypes were rescued. This important observation shows the independent impacts of a loss of GLI3 on testis biology as distinct from its role in the development of the external genitalia and anogenital spacing.

This is a well written paper with well designed experiments and a thorough investigation of the role of Gli3 in the male repro tract. Some of the pictures were low-res [no doubt due to pdf conversation] which detracted from some of the beautiful whole mount images.

The only question I was left wondering was if you had looked at downstream HH targets in the GT by qPCR to confirm your hypothesis there?

Minor corrections:

Line 285 – ‘urethral tube defects associated with hypospadias’ – wording is strange here, urethral plate development was not examined, better to just say ‘caused hypospadias’

Line 301 ‘reproductive portion of the urogenital tract and the anorectal region’ here the author is referring to the external genitalia and the anal region. Better to state that clearly – the current wording is confusing, especially the use of ‘anorectal’.

Would be interesting to examine hh signalling in the GT [via qPCR] to confirm the conclusion that urethral defect is a direct result of disrupted hedgehog signaling in the genital tubercle

Reviewer #3: This is to elucidate function of GLI3 during testicular development in mice. Using Gli3XtJ mutant mouse line, spontaneous and semi-dominant mutation, the authors find reduction of Leydig cells in the mutants leading to decline of testicular hormones. Androgen supplementation partially rescues some of abnormalities such as delay/lack of testicular descent but not hypospadias. Based on those results, the authors propose a novel function of activated form of GLI3 in Leydig cells as a regulator of male hormone productions.

This is a potentially interesting story to uncover new mechanisms for development of the male reproductive system. Data presented in each figure well support the authors’ conclusions. However, the analyses are superficial and observational for the current format. This reviewer would suggest some additional attempts to understand mechanistic features of the abnormalities found in the mutant mice.

1. At the several places including the author summary, the authors state there is an interaction between the Hedgehog and androgen signaling pathways. What are the data to support this argument? Data shown in Fig 2B suggest that the production of testosterone is regulated by Hedgehog signaling activity (or other GLI3 activities), which is different from the idea of interaction.

2. For Fig 1J, it would be helpful to show gross photos to indicate how measurements were done to get ADG. Please provide information which stage(s) the authors made this observation (seemingly at E16.5, but description could be much clearer). Did the authors check genders? If so, how? By checking structures of the reproductive tracts? Presence/absence of Sry? Are there any possibilities that the Gil3XtJ mutation causes sex reversal?

3. What is known about the mechanisms of how transcription of Insl3 and/or production of testosterone are regulated by Hedgehog signaling or other functions of activated GLI3? Is there any information to speculate this is a direct interaction or secondary? Data shown in Fig. 2AB suggest that GLI3 is not an essential component for production of testicular hormones, but plays a role to augment productions. What is known about the mechanisms of stage-dependent massive increases of testicular hormones and what is the authors’ speculation how GLI3 would be involved in this mechanism?

4. Data shown in Fig 3 are intriguing. They convincingly depict different behaviors of cranial suspensory ligament (CSL), which is testosterone-dependent, and caudal gubernaculum ligament, which is not. Similar to above (#3), what is known about the mechanisms of regression of CSL and how/which step exogenous DHT would rescue persistence of fibers in the ligament?

5. It is reported that increased levels of MIS can cause failure of testicular descent. Although it is not quantitative, the image shown in Fig 4H&I may prompt to speculate one of the possible reasons for the delay of testicular descent is over produced MIS.

**Have all data underlying the figures and results presented in the manuscript been provided?**

Reviewer #1: Yes

Reviewer #2: Yes

Reviewer #3: Yes

PLOS authors have the option to publish the peer review history of their article (what does this mean?). If published, this will include your full peer review and any attached files.

Reviewer #1: No

Reviewer #2: No

Reviewer #3: Yes: Yuji Mishina

---

## [Decision Letter · Decision Letter 1]

28 Apr 2020

Dear Dr Jorgensen,

We are pleased to inform you that your manuscript entitled "GLI3 Resides at the Intersection of Hedgehog and Androgen Action to Promote Male Sex Differentiation" has been editorially accepted for publication in PLOS Genetics. Congratulations!

Yours sincerely,

Moira K. O'Bryan

Associate Editor

PLOS Genetics

Scott Williams

Section Editor: Natural Variation

PLOS Genetics

Comments from the reviewers (if applicable):

This manuscript is now ready for publication. Congratulations.

Reviewer's Responses to Questions

**Comments to the Authors:**

Reviewer #1: The authors have addressed my few concerns in the revised manuscript.

Reviewer #3: The authors intensively and appropriately responded the comments/suggestions raised by this reviewer and others. Addition of figure 8 significantly increases the visibility of the authors' achievements. Revised text is highly logistic and enjoyable.

**Have all data underlying the figures and results presented in the manuscript been provided?**

Reviewer #1: Yes

Reviewer #3: Yes

PLOS authors have the option to publish the peer review history of their article (what does this mean?). If published, this will include your full peer review and any attached files.

Reviewer #1: No

Reviewer #3: Yes: Yuji Mishina

**Data Deposition**

http://datadryad.org/submit?journalID=pgenetics&manu=PGENETICS-D-20-00050R1

**Press Queries**

---

## [Editor Report · Acceptance letter]

29 May 2020

PGENETICS-D-20-00050R1 

GLI3 Resides at the Intersection of Hedgehog and Androgen Action to Promote Male Sex Differentiation 

Dear Dr Jorgensen, 

We are pleased to inform you that your manuscript entitled "GLI3 Resides at the Intersection of Hedgehog and Androgen Action to Promote Male Sex Differentiation" has been formally accepted for publication in PLOS Genetics! Your manuscript is now with our production department and you will be notified of the publication date in due course.

With kind regards,

Kaitlin Butler

PLOS Genetics

On behalf of:
